# Partial cross mapping eliminates indirect causal influences

Siyang Leng [1,2,3], Huanfei Ma [4], Jürgen Kurths[5,6], Ying-Cheng Lai [7], Wei Lin [1,2,8 ✉],
Kazuyuki Aihara [3,9 ✉] & Luonan Chen [10,11,12,13 ✉]

Causality detection likely misidentifies indirect causations as direct ones, due to the effect of causation transitivity. Although several methods in traditional frameworks have been proposed to avoid such misinterpretations, there still is a lack of feasible methods for identifying direct causations from indirect ones in the challenging situation where the variables of the underlying dynamical system are non-separable and weakly or moderately interacting. Here, we solve this problem by developing a data-based, model-independent method of partial cross mapping based on an articulated integration of three tools from nonlinear dynamics and statistics: phase-space reconstruction, mutual cross mapping, and partial correlation. We demonstrate our method by using data from different representative models and real-world systems. As direct causations are keys to the fundamental underpinnings of a variety of complex dynamics, we anticipate our method to be indispensable in unlocking and deciphering the inner mechanisms of real systems in diverse disciplines from data.

[1] School of Mathematical Sciences, SCMS, SCAM, and LMNS, Fudan University, 200433 Shanghai, China. [2] Center for Computational Systems Biology of ISTBI, LCNBI, and Research Institute of Intelligent Complex Systems, Fudan University, 200433 Shanghai, China. [3] Institute of Industrial Science, University of Tokyo, Tokyo 153-8505, Japan. [4] School of Mathematical Sciences, Soochow University, 215006 Suzhou, China. [5] Potsdam Institute for Climate Impact Research, Potsdam 14412, Germany. [6] Saratov State University, Saratov 410012, Russia. [7] School of Electrical, Computer, and Energy Engineering, Arizona State University, Tempe, AZ 85287-5706, USA. [8] State Key Laboratory of Medical Neurobiology, and MOE Frontiers Center for Brain Science, Institutes of Brain Science, Fudan University, 200032 Shanghai, China. [9] International Research Center for Neurointelligence (IRCN), University of Tokyo, Tokyo 113-0033, Japan. [10] Center for Excellence in Molecular Cell Science, Institute of Biochemistry and Cell Biology, Chinese Academy of Sciences, 200031 Shanghai, China. [11] Center for Excellence in Animal Evolution and Genetics, Chinese Academy of Sciences, 650223 Kunming, China. [12] Institute of Brain-Intelligence Technology, Zhangjiang Laboratory, 201210 Shanghai, China. [13] Key Laboratory of Systems Biology, Hangzhou Institute for Advanced Study, University of Chinese Academy of Sciences, Chinese Academy of Sciences, 310024 Hangzhou, China. ✉email: wlin@fudan.edu.cn; aihara@sat.t.u-tokyo.ac.jp; lnchen@sibs.ac.cn

Causal interactions are fundamental underpinnings in natural and engineering systems, as well as in social, economical, and political systems. Here system details are typically not known, but only time series are available. Correctly identifying causal relations among the dynamical variables generating the time series provides a window through which the inner dynamics of the target system may be probed into, and a number of previous methods were developed, such as those based on the celebrated Granger causality[1–5], the entropy[6–11], the dynamical Bayesian inference[12–15], and the mutual cross mapping (MCM)[16–21], with applications to real-world systems[5,7,22–31]. If the system contains two independent variables only, the causal relation between them is straightforwardly direct. However, for a complex system with a large number of interacting nodes connected with each other in a networked fashion, two kinds of causation can arise: direct and indirect. Especially, if there is a direct link between two nodes, the detected causal relation between them can contain a direct component and an indirect one through other nodes in the network as a result of the generic phenomenon of causation transitivity (see Fig. 1). Even for two nodes that are not directly connected, a causal relation may be detected, but it must be indirect. To eliminate indirect causal influences so as to ascertain direct causal links is of paramount importance, as the latter constitutes the base for modeling, predicting, and controlling the system. There were previous studies of significant advance in detecting direct causal links to reconstruct the underlying true causal network based on the concept of partial transfer entropy or its linear Gaussian version, the conditional Granger causality, which resulted in many successful data mining in related fields[32–38]. Combining these methods with graphical models, recent studies further provided a visible and comprehensive description of causal relations among interested variables[36,38,39]. However, mathematically, all these methods are not applicable directly in situations where the relevant dynamical variables are non-separable so that the information from any variables cannot be separated easily in a prediction framework (see "Methods" for the rigorous concept of

non-separability). In real-world nonlinear systems, the non-separability is ubiquitously present among systems variables[17]. To our knowledge, the problem of ascertaining direct causation by removing indirect causal influences for general complex dynamical systems has not been fully studied and remained outstanding.

In this paper, we develop a data-based, model-free method of partial cross mapping (PCM) to eliminate indirect causal influences in situations where non-separability is allowed to be present. The central idea is to integrate three basic data analysis methods from nonlinear dynamics and statistics: classic phase-space reconstruction, MCM, and partial correlation, to detect direct causal links for complex and nonlinear networked systems. The method is validated using various benchmark systems. Its applications to real-world systems lead to new insights into their dynamical underpinnings. The method provides a solution to the long-standing, crucial problem with existing causality detection methods: misidentifying indirect causal influences as direct ones. Because of its unprecedented ability to eliminate indirect causation, this method can be a powerful tool to understand and model complex dynamical systems.

## Results

**Direct and indirect causal links**. To illustrate the difference between direct and indirect causal links, we first consider a toy system of three variables with different interaction structures. If only two variables interact in one direction and the third one is isolated (Fig. 1a), then the previous methods can be effective for identifying the direct causal link[16–21]. However, when the three variables constitute a unidirectional causal chain (Fig. 1b), applying any of the previous methods to the time series from a pair of variables would detect a false direct link between the two non-neighboring variables $X$ and $Y$ in Fig. 1b (see "Methods" for a false link aroused by the transitivity). When the three variables constitute a causal loop (Fig. 1c), every two neighboring variables may have an indirect causal link in addition to the direct one in the opposite direction. In this case, previous methods would falsely identify any actual indirect link as a direct one. In addition to the above three representative interaction structures for the three variables, all the other possible modes have been introduced thoroughly and investigated systematically in Supplementary Note 1. Moreover, with more observable variables, the likelihood that indirect causal links are incorrectly regarded as direct ones will substantially increase (Fig. 1d).

**Partial cross mapping**. To overcome this problem, we propose the PCM method. The key idea is to examine the consensus between one time series and its cross map prediction from the other with conditioning on the part that is transferred from the third variable. For the convenience of describing our method clearly, we consider the simple case of three variables ($X$, $Y$, and $Z$) causally interacting with each other in a unidirectional chain (Fig. 2a). Let $X = \{x_t\}_{t=1}^L$, $Y = \{y_t\}_{t=1}^L$, and $Z = \{z_t\}_{t=1}^L$ be the corresponding time series of length $L$. Using Takens–Mañé's delay-coordinate embedding[40,41], we obtain three shadow manifolds: $M_X = \{\mathbf{x}_t\}_{t=r}^L$, $M_Y = \{\mathbf{y}_t\}_{t=r}^L$, and $M_Z = \{\mathbf{z}_t\}_{t=r}^L$ with the vectors

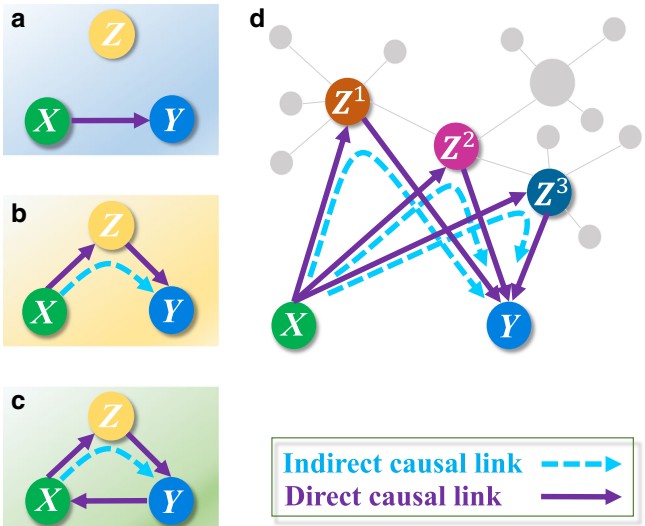

**Fig. 1 Direct versus indirect causal links. a** There is directional interaction between variables $X$ and $Y$, but $Z$ is an independent variable. **b** The variables $X$, $Y$, and $Z$ constitute a one-directional causal chain with an indirect causal link from $X$ to $Y$. **c** The variables constitute a causal loop, where every two neighboring variables have, in two opposite directions, a direct and an indirect causal link, respectively. **d** For a network with many interacting variables, more indirect causal links would be falsely identified as direct causal links.

Indirect causal link ---->
Direct causal link ——>

$$\mathbf{x}_t = (x_t, x_{t-\tau_x}, \dots, x_{t-(E_x-1)\tau_x}),$$
$$\mathbf{y}_t = (y_t, y_{t-\tau_y}, \dots, y_{t-(E_y-1)\tau_y}),$$
$$\mathbf{z}_t = (z_t, z_{t-\tau_z}, \dots, z_{t-(E_z-1)\tau_z}),$$

where $E_x$, $E_y$, and $E_z$ are the respective embedding dimensions, $\tau_x$, $\tau_y$, and $\tau_z$ are the time lags, and $r = \max_{\xi=x,y,z}\{1 + (E_\xi - 1)\tau_\xi\}$.

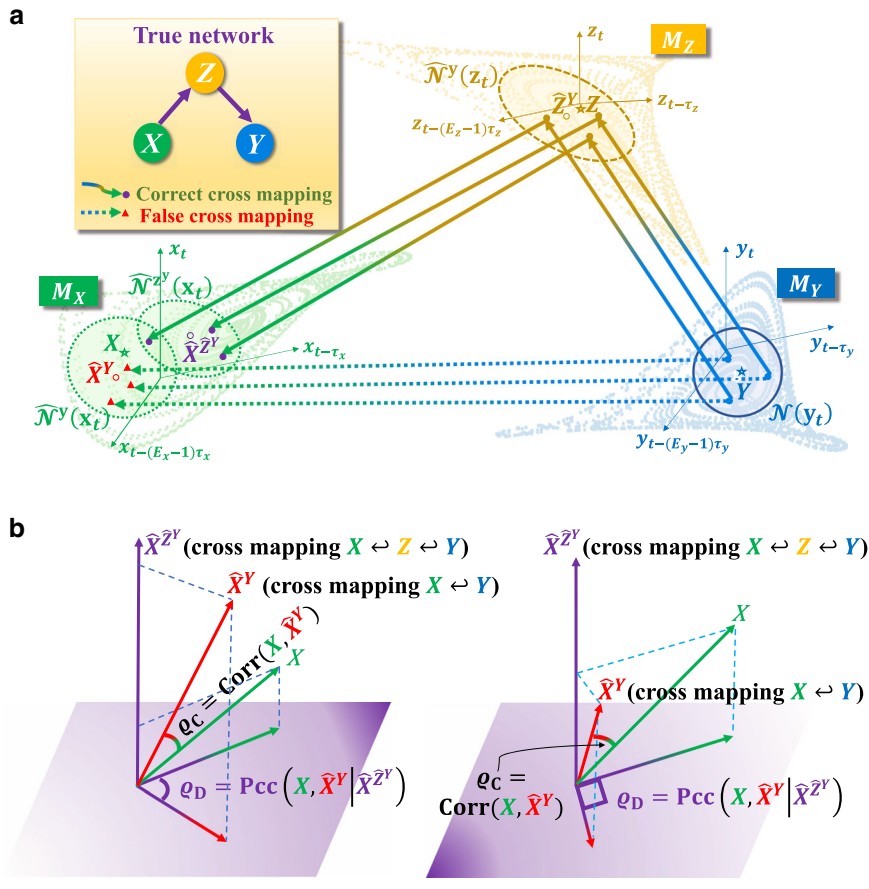

**Fig. 2 Basic principles of the PCM framework. a** For the illustrative setting of three variables interacting in a unidirectional causal chain, the MCM method maps $\mathcal{N}(\mathbf{y}_t)$ to the left circled region $\hat{\mathcal{N}}^Y(\mathbf{x}_t)$ in $M_X$, where the estimated $\hat{X}^Y$ is close to the true $X$, denoting full causal information from $X$ to $Y$ and leading to erroneous identification of the indirect causal link as the direct link. **b** For our proposed PCM method, partial correlation coefficient between $X$ and $\hat{X}^Y$ is calculated by conditioning on the information about $\hat{X}^{\hat{Z}^Y}$, which is mapped from $\mathcal{N}(\mathbf{y}_t)$ through $M_Z$ and then to the right circled region $\hat{\mathcal{N}}^{Z^Y}(\mathbf{x}_t)$ in $M_X$ in **a**, denoting indirect information flow. Geometrically, $\varrho_C$ corresponds to the cosine of the angle between $X$ and $\hat{X}^Y$ in the entire space, while $\varrho_D$ is the projection of $\varrho_C$ onto the subspace orthogonal to $\hat{X}^{\hat{Z}^Y}$. Because $\varrho_C \geq \varrho_D$, the example in **a** corresponds to the sketch on the right side of **b**, where the projection is close to the right angle, leading to a near-zero value of $\varrho_D$.

These parameters of embedding dimensions and time lags can be computationally determined by the method of false nearest neighbor (FNN) and delayed mutual information (DMI), respectively. More advanced techniques can also be utilized[20,42]. In general, for any pair of variables $\xi$ and $\eta \in \{\mathbf{x}, \mathbf{y}, \mathbf{z}\}$, we set $\hat{\mathcal{N}}^\xi(\eta_t) = \{\eta_{t'}|\xi_{t'} \in \mathcal{N}(\xi_t)\}$, where $\mathcal{N}(\xi_t)$ is a set containing a fixed number (usually taken as $E_\xi + 1$, which is the minimum number of points needed for a bounded simplex in an $E_\xi$-dimensional space[43]) of nearest neighboring points of $\xi_t$ in the corresponding shadow manifold. For $\xi = \eta$, $\hat{\mathcal{N}}^\xi(\eta_t)$ becomes $\mathcal{N}(\eta_t)$. For $\xi \neq \eta$, $\hat{\mathcal{N}}^\xi(\eta_t)$ becomes a cross mapping neighborhood from $\mathcal{N}(\xi_t)$ (for an illustrative example, see the horizontal arrows from $M_Y$ to $M_X$ in Fig. 2a). The dependence from $\mathcal{N}(\eta_t)$ to $\hat{\mathcal{N}}^\xi(\eta_t)$ characterizes the causal influence from the variable producing $\eta_t$ to the variable producing $\xi_t$. Previously developed heuristic measures for quantifying such dependence and causal influence[16–18,20,21] constitute the MCM framework. We exploit the correlation coefficient[17] between $\eta_t$ and $\hat{\eta}_t^\xi = \mathbb{E}[\hat{\mathcal{N}}^\xi(\eta_t)]$, where $\hat{\eta}_t^\xi$ is the mapping from $\xi_t$ and $\mathbb{E}[\cdot]$ is an operation taking appropriately weighted average over all the points in a given set.

Specifically, if the correlation coefficient $\varrho_C = |\text{Corr}(\mathbf{x}_t, \hat{\mathbf{x}}_t^Y)|$ is larger than an empirical threshold $T$, the MCM method will stipulate that there is a causal influence from $X$ to $Y$. MCM complements the field of causality analysis in pairwise non-separable dynamical systems. However, due to causation transitivity, the causal link detected by MCM can be either direct or indirect, as illustrated in Fig. 2a. Additionally, since causation manifests its influence in a certain time delay, we search for an optimal time delay that maximizes the causation (i.e., the obtained correlation coefficient $\varrho_C$) between a translated $Y$ and $X$ (see "Methods" for a detailed description)[20].

Heuristically, $\varrho_C$, as defined above, represents the cosine of the angle between $X$ and $\hat{X}^Y$ in the entire space, as shown in Fig. 2b. In order to distinguish the existence of the causation transitivity, we consider the projection of $\varrho_C$ onto the information space orthogonal to the indirect information that is induced by the causation transitivity. To this end, we formulate our PCM framework (see "Methods" and Supplementary Fig. 1 for detailed formulations and practical instructions). First, for a time series pair $Z$ and translated $Y_{\tau_i} = \{y_{t+\tau_i}\}$ with possible time delay candidates $\tau_i(i = 1, 2, \dots, m)$, we apply the conventional MCM method to determine the optimal time delay $\tau_i = \tau_{i_1}$, which

maximizes the correlation coefficient $\text{Corr}(Z, \hat{Z}^{Y_{\tau_i}})$. Correspondingly, the obtained mapping $\hat{Z}^{Y_{\tau_{i_1}}}$ from $Y_{\tau_{i_1}}$ is denoted by $\hat{Z}^Y$ for simplicity. The next step is to repeat the procedure to the time series pair $X$ and the translated $\hat{Z}^Y_{\tau_i}$ so as to obtain the optimal time delay $\tau_{i_2}$, as well as the mapping $\hat{X}^{\hat{Z}^Y_{\tau_{i_2}}}$ from $\hat{Z}^Y_{\tau_{i_2}}$, which maximizes the coefficient $\text{Corr}(X, \hat{X}^{\hat{Z}^Y_{\tau_i}})$. Denoting the obtained mapping by $\hat{X}^{\hat{Z}^Y}$, which is acquired from a successive MCM procedure and characterizes the indirect information flow through $Z$, and then obtaining $\hat{X}^Y$, which characterizes all causal information from $X$ to $Y$, by repeating the above procedure to time series pair $X$ and the translated $Y_{\tau_i}$, we introduce the correlation index: $\varrho_D = \left| \text{Pcc}(X, \hat{X}^Y | \hat{X}^{\hat{Z}^Y}) \right|$ to measure the direct causation from $X$ to $Y$ conditioned on the indirect causation through $Z$, where $\text{Pcc}(\cdot, \cdot | \cdot)$ is the partial correlation coefficient describing the association degree between the first two variables with information about the third variable removed[44], in contrast to the MCM index $\varrho_C = \left| \text{Corr}(X, \hat{X}^Y) \right|$. Note that we search for the strongest causation on different candidate time delays in every MCM procedure above. As a consequence, $\varrho_D$ can be regarded intuitively as the projection of $\varrho_C$ onto the information space orthogonal to the indirect information $\hat{X}^{\hat{Z}^Y}$ (Fig. 2b), and thus eliminates the indirect causal influence.

For three causally interacting variables $X$, $Y$, and $Z$, we generally have $\varrho_C \geq \varrho_D$. Setting an empirical threshold $1 > T \gg 0$, we have three cases for the order of the correlation index: $\varrho_C \geq \varrho_D \geq T$, $\varrho_C \geq T \gg \varrho_D$, and $T > \varrho_C \geq \varrho_D$, corresponding, respectively, to the three causal relations: a direct causal link from $X$ to $Y$, a sole indirect causal link from $X$ to $Y$, and the absence of any causal link from $X$ to $Y$. The index $\varrho_D$ thus characterizes the degree to which direct causal links can be ascertained while eliminating the possibility of indirect links. For the example in Fig. 2a, the causal interaction of $X$ and $Y$ belongs to the second case above, which can be inferred from the correlation index in the same order as $\varrho_C \geq T \gg \varrho_D$. In real applications, it can happen that the causal signals in transition are not strong enough, making the values of $\varrho_C \gtrsim T$ and $\varrho_D$ close to that of $T$. In such a case, the detection of direct causal links becomes more sensitive to the value of $T$. To overcome this difficulty, we introduce $\gamma = \varrho_D/\varrho_C$ to measure the proximity of the two index values. The closer the proximity to one, the higher the possibility of the existence of a direct causal link. Multiple tests[45–47] have been conducted to ensure statistical reliability.

The framework of PCM can be generalized to networked systems with an arbitrary number of interacting variables: $X$, $Y$, $Z^1$, ..., $Z^s$ ($s \geq 2$) (e.g., Fig. 1d). With the full correlation between $X$ and $\hat{X}^Y$, we calculate their partial correlation coefficient, denoted as $\varrho_{D_1} = \left| \text{Pcc}(X, \hat{X}^Y | \{\hat{X}^{\hat{Z}^{iY}} | i = 1, \ldots, s\}) \right|$, by removing the information of the cross mapping variables from the $s$ variables $Z^1$, ..., $Z^s$, where $\varrho_{D_1}$ is a first-order measure for distinguishing the direct from indirect causal link from $X$ to $Y$. Motivation and formalization for extending this measure to higher orders is described in "Methods" section. We emphasize here that strongly coupled (synchronized) variables in nonlinear systems are not in the scope of the PCM framework, because in this circumstance the complete system collapses to the cause system sub-manifold, and the effect variable becomes an observation function on the cause system, where bidirectional causation will always be computationally detected[17]. In addition, theoretically our PCM framework is based on the Takens–Mañé theorem, which is applicable only for autonomous systems. Data entirely recorded from nonautonomous systems are therefore not directly suitable for this framework[48], but our method can be applied to some nonautonomous systems. In particular, it can be numerically used to detect piecewise causations with data from switching systems where the switching points could be located and each duration between the consecutive switching points is sufficiently long. Also, our framework is suitable for some forced systems or/and systems with weak or moderate noise because some generalized embedding theorems could support the soundness of our framework[49,50]. As for an important kind of nonautonomous system, viz., dynamical oscillators with time-evolving coupled functions or/and with various types of noise, the dynamical Bayesian inference with a delicate set of function bases can provide pretty practical solutions[14]. As for the future research topics, possible investigations include combining the above mutually complementary methods for causation detection in more general dynamical systems without knowing explicit model equations but with highly complex interaction structures.

**Ascertaining direct causation in benchmark systems**. To validate our PCM method, we use the following benchmark system of three interacting species: $x_t = x_{t-1}(\alpha_x - \alpha_x x_{t-1} - \beta_{xy} y_{t-1}) + \epsilon_{x,t}$, $y_t = y_{t-1}(\alpha_y - \alpha_y y_{t-1} - \beta_{yx} x_{t-1} - \beta_{yz} z_{t-1}) + \epsilon_{y,t}$, and $z_t = z_{t-1}(\alpha_z - \alpha_z z_{t-1} - \beta_{zx} x_{t-1}) + \epsilon_{z,t}$, for $\alpha_x = 3.6$, $\alpha_y = 3.72$, and $\alpha_z = 3.68$, where $\epsilon_{i,t}$ ($i \in \{x, y, z\}$) are white noise of zero mean and standard deviation 0.005. Different choices of the coupling parameters $\beta_{xy}$, $\beta_{yx}$, $\beta_{yz}$, and $\beta_{zx}$ can lead to distinct interacting modes (Fig. 3a). From the time series, we compute the MCM and PCM indices, $\varrho_C$ and $\varrho_D$, respectively, for detecting the causal link from $X$ to $Y$, with results listed in Fig. 3b, c. While there are cases where both methods are effective at detecting the direct causal links, for the causal chain and the causal loop structures with the threshold value $T = 0.5$, the PCM method succeeds in discriminating the indirect causal links, while clearly the MCM method, without eliminating the influence of the causation transitivity, fails. As furher shown in Supplementary Note 2, the PCM performance is more robust than that of the MCM method with respect to variations in the value of $T$, making the PCM method applicable to real-world systems when there is none or little a priori knowledge of assigning a proper value of $T$. The results in Fig. 3b, c have also been verified by using the multi-testing corrections. Additionally, for all the other possible interaction structures of three species, including the representative network motifs: fan-in, fan-out, and cascading structures[51,52], our systematic studies manifest that the PCM method achieves accurate causation detections completely (see Supplementary Note 1). More importantly, we systematically conducted comparison studies with the Granger causality, the transfer entropy and all their conditional extensions to detect the causations for the above three species system and tested their robustness against different noise levels and time series lengths. As clearly shown in Supplementary Note 3, the PCM outperforms those existing methods which are, in principle, suitable only for the variables satisfying the separability condition. We also provided a comparison study between the PCM framework and the dynamical Bayesian inference in Supplementary Note 3. Both methods have their own particular advantages and could be used in a complementary manner. All these results systematically demonstrate the universal and peculiar usefulness of our method to the typical situation where the variables of dynamical systems are non-separable.

Additionally, we validate the effectiveness of the PCM method in a network model containing eight interacting species. As

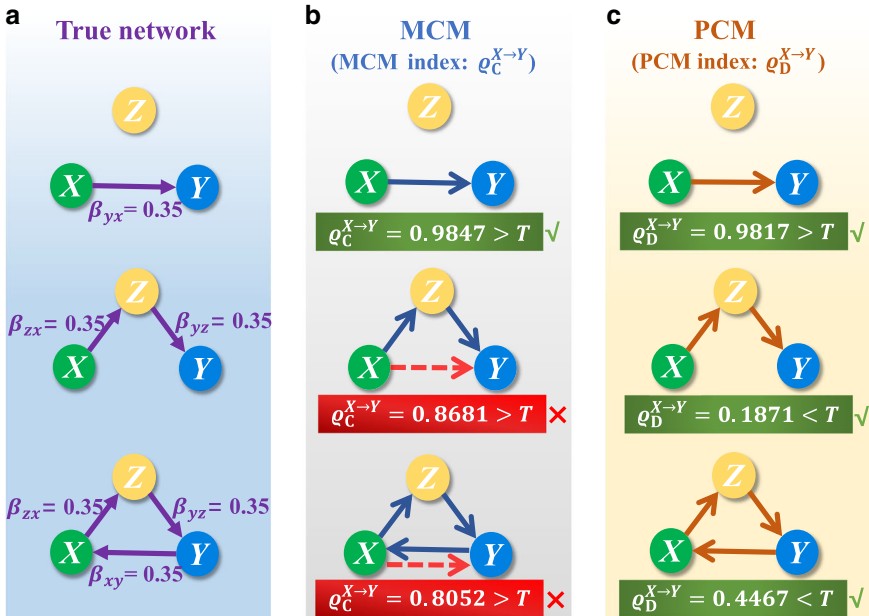

**Fig. 3 Detection of causal links from X to Y in the benchmark systems. a** Three distinct interaction modes of the system. **b** Causal links from $X$ to $Y$ detected by the MCM method, which contain false direct causation for the second and the third interaction modes. **c** Direct causal links detected by the PCM method, which successfully excludes the false direct causations in **b**. Randomly selected are the 100 trials with a 1000-length from 5000-length time series, where the sampling rate is 1 Hz so that the length matches exactly the time unit of the system. The average is calculated over the results of these randomly selected trials. The phase-space reconstruction parameters are $E = 4$ and $\tau = 1$. Here superscripts of $\varrho_C$ and $\varrho_D$ denote the specified causal direction.

shown in Supplementary Fig. 10, the direct causal network can be reconstructed faithfully while the indirect links are all eliminated successfully with setting an appropriate group of $T$. In contrast, with the same values of $T$, the MCM method produces a dense network containing direct, indirect, and even erroneous causal links. We also find that the ratio $\gamma = \varrho_D/\varrho_C$ can be used to improve the detection accuracy even for relatively small values of the threshold $T$ (Supplementary Note 4). Moreover, selecting a practically effective threshold value is much more realizable and robust in our PCM method (see Supplementary Fig. 11 and see Supplementary Note 5 for detailed information on statistical tests and methods for threshold selection). The robustness tests of PCM against the time series lengths and the noise scales also show good effectiveness even with small data size and relatively strong noise in this model (Supplementary Note 3). These additional results demonstrate the power of our PCM method in detecting direct links and accurately reconstructing the underlying causal networks from multivariate time series.

**Detecting direct causation in real-world networks.** We test gene regulatory networks with gene expression data available from DREAM4 in silico Network Challenge[53–55]. There are five networks with different, synthetically produced structures. Each network has 100 genes. We use the software GeneNetWeaver[56] to randomly select 20 interacting genes, where each gene has 10 realizations of 21 gene expression time series data. Figure 4a presents one gene regulatory network (see Supplementary Fig. 12 for the others). For each gene, we combine all realizations as one time series for phase-space reconstruction. We compare the direct causal links detected by PCM with the a priori known edges of the five networks and calculate the respective ROC (receiver operating characteristic) curves (Fig. 4b). We find the average of the five areas under the ROC curves approaches the value of ~0.75, indicating high detection accuracies of direct links in gene

regulatory networks even with small data sets, a task for which PCM outperforms the MCM method (see Supplementary Note 6).

We next consider the food chain network of three plankton species: *Pico cyanobacteria*, *Rotifers* and *Cyclopoids*, with the prey–predator relations indicated in Fig. 4c. The oscillatory population data are selected from an 8-year mesocosm experiment of a plankton community isolated from the Baltic Sea[57–59]. Our PCM method yields six indices for all the possible causal links, and we preserve the links with index values $\gtrsim 10^{-1}$ and discard other links (see Supplementary Note 5 for issues on threshold selection). This leads to two direct causal links, which agree with the ground truth of the original network (Fig. 4d). Remarkably, our PCM method successfully excludes the indirect link from *Pico cyanobacteria* to *Cyclopoids*. For this network, there is also a weak direct link from *Rotifers* to *Pico cyanobacteria*, and our method is indeed able to detect it (verified with multi-testing corrections). This reveals that the actual prey–predator hierarchy does not necessarily match the direct causal links among the species. For example, while predators hunt preys, a predator through hunting can significantly influence the prey populations when they are not tremendously abundant. In such a case, the predator can be regarded as the causal source, giving rise to the third relatively weak but direct causal link.

Our third real-world example is from the recorded data of air pollution and hospital admission of cardiovascular diseases in Hong Kong from 1994 to 1997 (see Supplementary Note 6)[60–62]. As shown in Fig. 4e, f, our PCM method uncovers that only the pollutants, that is, nitrogen dioxide and respirable suspended, are detected as the major causes of cardiovascular diseases. Neither sulfur dioxide nor ozone has been identified as the cause for the diseases, which is consistent with previous results[20,63]. Our method reveals a unidirectional causal relation from ozone to sulfur dioxide, but the detected causal relations among the recognized pollutants are bidirectional. It is likely that these detected causal relations are either direct or indirect, because data

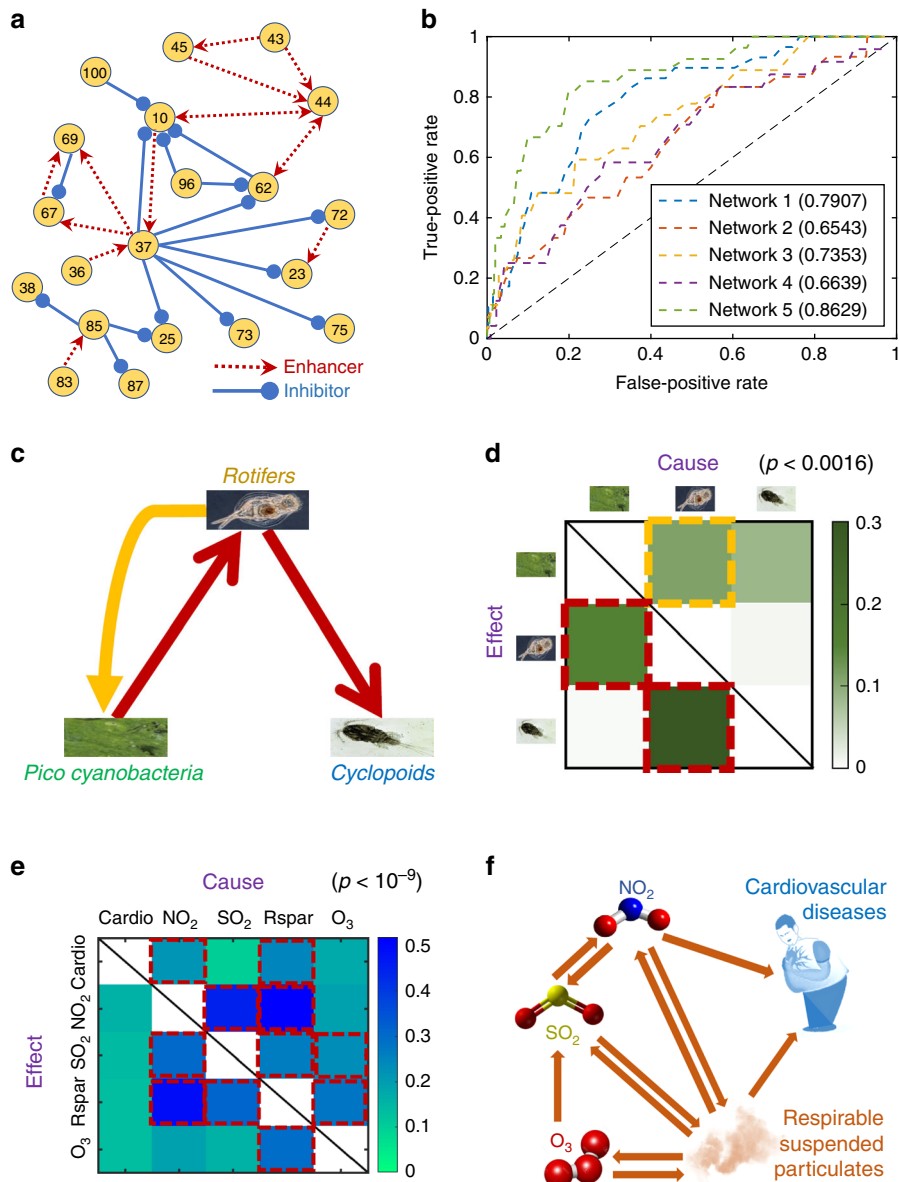

**Fig. 4 Detecting direct causal links in three real-world networks. a** One of the five gene regulatory networks with 20 interacting genes from GeneNetWeaver. Each red (blue) arrow represents an activating (inhibitory) effect. **b** ROC curves characterizing the PCM detection performance. The corresponding AUROCs are also indicated. The reconstruction parameters are $E = 2$ and $\tau = 1$. **c** A food chain network of three plankton species, where the direction of each red arrow represents a prey to predator interaction. **d** The PCM indices (the color region framed by red boxes) signifying successful detection of the direct causal links (for $E = 4$ and $\tau = 1$). A relatively weak but direct causal link (the yellow arrow in **c**) from *Rotifers* to *Pico cyanobacteria* is identified through the index framed by the yellow box. **e** Results on all successfully detected interactions between air pollutants and cardiovascular diseases (red box) for $E = 7$ and $\tau = 1$. **f** The reconstructed causal network from the results in **e**. All detection results are verified using multiple testing corrections.

of other factors, such as temperature, humidity, and wind speed, are not completely available, which can be the common causes to some pollutants (e.g., the fan-out interaction mode shown in Supplementary Fig. 2).

We also apply the PCM method to real-world examples, including gene expression data related to the circadian rhythms and electroencephalography data of the human brain in Supplementary Note 7. All the results demonstrate the broad applicability of our method to different scales of data sets, and indeed reveal new viewpoints to the dynamical underpinnings of real-world systems.

## Discussion

To summarize the work, by exploiting both dynamical and statistical features from the observed data, there are two major

advantages of our method: detecting direct causality based on PCM and handling non-separability problem based on Takens–Mañé's embedding theorem. Actually, variables for a nonlinear dynamical system are generally considered non-separable due to their intertwined nonlinear nature. Specifically, in contrast to the existing methods on detecting causation, which either misidentify indirect causal links as direct ones or fail due to a violation of the condition of separability, we develop a method theoretically and computationally to solve this outstanding problem, coping with the situation for which the existing frameworks cannot work effectively. The central idea lies in examining the consensus between one time series and its cross map prediction from the other with conditioning on the part that is transferred from the third variable. Our method is capable of not only distinguishing direct from indirect causal

influences but also removing the latter. A virtue of our method is that it is generally applicable to nonlinear dynamical networks without requiring the condition of separability, which complements the missing part of causality analysis (see Supplementary Table 3). In fact, the concept of causality in dynamical systems is different from the widely accepted traditional statistical viewpoint that $X$ causes $Y$ if and only if an intervention in $X$ has an effect on $Y$. Due to the non-separability, causality in dynamical systems should have different formalization, which in simplest way can be intuitively interpreted as a coupling term from $X$ to $Y$ in the system's equations. Further theoretical interpretations regarding this new framework will be included in our future work. Finally, our PCM method is validated by applying to a number of real-world systems, yielding new insights into the dynamics of these systems. Unambiguous identification of direct causal links with indirect causal influence eliminated is a key to understanding and accurately modeling the underlying system, and our framework therefore provides a vehicle to achieve this goal.

## Methods

**The concept of non-separability.** We illustrate the concept, non-separability, by using a general continuous-time dynamical system:

$$\dot{\mathbf{x}} = F(\mathbf{x}), \tag{1}$$

where the state variable $\mathbf{x}(t) = [x_1(t), x_2(t), \ldots, x_n(t)]^\top$ evolves inside a compact manifold $\mathcal{M}_x$, forming an attractor $\mathcal{A}$ with a dimension $d_\mathcal{A}$. Here, $d_\mathcal{A}$ can be computed as the box-counting dimension of $\mathcal{A}$. The dynamics with an initial value $\mathbf{x}_0 \in \mathcal{M}_x$ are denoted by $\mathbf{x}(t) = \varphi_t(\mathbf{x}_0)$, where $\varphi_t(\cdot)$ is regarded as a flow along the manifold $\mathcal{M}_x$. According to Takens–Mañé's embedding theory and its fractal generalizations, one can, with probability one, reconstruct the system with a positive delay $\tau$ and a smooth observation function $h : \mathcal{M}_x \to \mathbb{R}$ in the sense that the delay-coordinate map $\Gamma_{h,\varphi,\tau}(\mathbf{x}) = [h(\mathbf{x}), h(\varphi_{-\tau}(\mathbf{x})), h(\varphi_{-2\tau}(\mathbf{x})), \ldots, h(\varphi_{-(L-1)\tau}(\mathbf{x}))]^\top$ is generically an embedding map as long as $L > 2d_\mathcal{A}$. Particularly for direct illustration, we take the observation function $h(\mathbf{x})$ as a simple coordinate function: $h(\mathbf{x}) = x_i$, where $x_i$ is the $i$th component of $\mathbf{x}$. Thus, we have $\mathbf{y}(t) = [x_i(t), x_i(t-\tau), \ldots, x_i(t-(L-1)\tau)]^\top$ and also have the manifold $\mathcal{M}_x$ mapped to the shadow manifold $\mathcal{M}_y$ by the embedding map $\Gamma$. Since the embedding map is one to one, the dynamics $\psi_\tau$ on the shadow manifold $\mathcal{M}_y$ are topologically conjugated with the dynamics $\varphi_\tau$ on $\mathcal{M}_x$, that is,

$$\mathbf{y}(t+\tau) = \psi_\tau(\mathbf{y}(t)) = \Gamma \circ \varphi_\tau \circ \Gamma^{-1}(\mathbf{y}(t)). \tag{2}$$

On the one hand, system (1) implies a fact that the future dynamics of one specific component, say $x_j$ with $j = (\text{or} \neq)i$, is governed by

$$[\varphi_\tau]_j : \mathbf{x}(t) = [x_1(t), x_2(t), \ldots, x_n(t)]^\top \to x_j(t+\tau) \tag{3}$$

and thus depends on the history of all the components $x_1, x_2, \ldots, x_n$. On the other hand, the relation in (2) implies the other fact that as long as the embedding map $\Gamma$ exists, the future dynamics of $x_j$ is also governed by

$$[\Gamma^{-1} \circ \psi_\tau]_j : \mathbf{y}(t) = [x_i(t), x_i(t-\tau), \ldots, x_i(t-(L-1)\tau)]^\top \to x_j(t+\tau) \tag{4}$$

and thus only depends on the history of one variable $x_i$ and on the embedding map $\Gamma$ as well.

Generically, it is possible to make a prediction of $x_j(t+\tau)$ based only on the observation of one variable, and this prediction could be as perfect as the prediction using the information of all the variables $x_1(t), x_2(t), \ldots, x_n(t)$ of the system (this obviously disables the idea of Granger causality and its extensions). Thus, Takens–Mañé's embedding theory reveals that, in such a deterministic nonlinear dynamical system, the information of the whole dynamical system could be generically injected into only one single variable and thus could be reconstructed by the observation data of that variable. This therefore invites a concept of non-separability, that is, one, prevalently, cannot remove the information of some variable from the other variables when any prediction is made for the dynamical systems. This also reveals that the methods based on prediction frameworks, such as the Granger causality, the transfer entropy, and all their extensions, mathematically are not suitable for dealing with the time series data produced by nonlinear dynamical systems where non-separability always exists among the internal variables. A toy example showing how GC fails in non-separable systems could be referred to the Supplementary Materials of ref. [17].

**Transitivity arousing indirect causation.** To investigate how the transitivity arouses indirect causation, we consider a heuristic logistic model of three species

connected in the following manner:

$$\begin{aligned} x_t &= x_{t-1}(\alpha_x - \alpha_x x_{t-1}), \\ z_t &= z_{t-1}(\alpha_z - \alpha_z z_{t-1} - \beta_{zx} x_{t-1}), \\ y_t &= y_{t-1}(\alpha_y - \alpha_y y_{t-1} - \beta_{yz} z_{t-1}), \end{aligned} \tag{5}$$

where the three species $X = \{x_t\}$, $Z = \{z_t\}$ and $Y = \{y_t\}$ are interacting in a causal chain, denoted by $X \to Z \to Y$, and the coupling strengths $\beta_{zx}$ and $\beta_{yz}$ are nonzero.

Now, we shift the second equation in (5) with one time step and then substitute it into the last equation in (5), which yields:

$$y_t = y_{t-1}\left[\alpha_y - \alpha_y y_{t-1} - \beta_{yz} z_{t-2}(\alpha_z - \alpha_z z_{t-2} - \beta_{zx} x_{t-2})\right]. \tag{6}$$

Also the last equation in (5) can be transformed as:

$$z_{t-1} = \frac{1}{\beta_{yz}}(\alpha_y - \alpha_y y_{t-1} - y_t/y_{t-1}), \tag{7}$$

so that

$$z_{t-2} = \frac{1}{\beta_{yz}}(\alpha_y - \alpha_y y_{t-2} - y_{t-1}/y_{t-2}). \tag{8}$$

Then, a substitution of Eq. (8) into Eq. (6) gives:

$$\begin{aligned} y_t = y_{t-1}\bigg\{ &\alpha_y - \alpha_y y_{t-1} - \beta_{yz}\frac{1}{\beta_{yz}}(\alpha_y - \alpha_y y_{t-2} - y_{t-1}/y_{t-2}) \\ &\times \left[\alpha_z - \alpha_z\frac{1}{\beta_{yz}}(\alpha_y - \alpha_y y_{t-2} - y_{t-1}/y_{t-2}) - \beta_{zx} x_{t-2}\right]\bigg\}. \end{aligned} \tag{9}$$

Consequently, this equation, coupling with the first equation in (5), forms a causation relation unidirectionally from $X$ to $Y$. However, this causation is indirect, induced by the transitivity, and then the influence has the effect of time delay for discrete-time dynamical systems.

**The PCM method of first order and higher order.** We now formulate the PCM framework formally (see Supplementary Fig. 1 for a schematic graph of the PCM procedure). The first step is to translate the time series $Y = \{y_t\}$ with time steps $\tau_i (i = 1, 2, \ldots, m)$, generating $m$ translated variables denoted as $Y_{\tau_i} = \{y_{t+\tau_i}\}$. For time series pair $Y_{\tau_i}$ and $Z$, we apply the conventional MCM method (see the practical steps below) to obtain the mapping $\hat{Z}^{Y_{\tau_i}}$ from $Y_{\tau_i}$ and calculate the correlation coefficient $\text{Corr}(Z, \hat{Z}^{Y_{\tau_i}})$. For simplicity, we denote $\hat{Z}^Y$ as the mapping $\hat{Z}^{Y_{\tau_{i_1}}}$ with

$$i_1 = \text{argmax}_{1 \le i \le m}\text{Corr}(Z, \hat{Z}^{Y_{\tau_i}}). \tag{10}$$

The next step is to repeat the procedure to the time series pair of translated $\hat{Z}^Y_{\tau_i}$ and $X$ so as to obtain the mapping $\hat{X}^{\hat{Z}^Y_{\tau_i}}$ from $\hat{Z}^Y_{\tau_i}$, and set $\hat{X}^{\hat{Z}^Y}$ as $\hat{X}^{\hat{Z}^Y_{\tau_{i_2}}}$ with

$$i_2 = \text{argmax}_{1 \le i \le m}\text{Corr}(X, \hat{X}^{\hat{Z}^Y_{\tau_i}}). \tag{11}$$

Now the obtained $\hat{X}^{\hat{Z}^Y}$ represents the indirect information flow. By directly applying MCM to the translated $Y_{\tau_i}$ and $X$, we could have $\hat{X}^Y$ denoting all the information transferred from $X$ to $Y$, which is simplified for $\hat{X}^{Y_{\tau_{i_3}}}$ with

$$i_3 = \text{argmax}_{1 \le i \le m}\text{Corr}(X, \hat{X}^{Y_{\tau_i}}). \tag{12}$$

We now introduce the correlation index:

$$\varrho_D = \left|\text{Pcc}(X, \hat{X}^Y | \hat{X}^{\hat{Z}^Y})\right|, \tag{13}$$

where $\text{Pcc}(\cdot, \cdot | \cdot)$ is the partial correlation coefficient describing the association degree between the first two variables with information about the third variable removed. We review the definition of partial correlation coefficient here. For time series $X$, $Y$, and $Z^1$, \ldots, $Z^s$, the partial correlation coefficient between $X$ and $Y$ conditioned on $Z^1$ is

$$\text{Pcc}(X, Y | Z^1) = \frac{\text{Corr}(X, Y) - \text{Corr}(X, Z^1)\text{Corr}(Y, Z^1)}{\sqrt{(1 - \text{Corr}(X, Z^1)^2)(1 - \text{Corr}(Y, Z^1)^2)}}. \tag{14}$$

The partial correlation coefficient between $X$ and $Y$ conditioned on both $Z^1$ and $Z^2$ is

$$\text{Pcc}(X, Y | Z^1, Z^2) = \frac{\text{Pcc}(X, Y | Z^1) - \text{Pcc}(X, Z^2 | Z^1)\text{Pcc}(Y, Z^2 | Z^1)}{\sqrt{(1 - \text{Pcc}(X, Z^2 | Z^1)^2)(1 - \text{Pcc}(Y, Z^2 | Z^1)^2)}}, \tag{15}$$

and the partial correlation coefficient between $X$ and $Y$ conditioned on more variables can be defined recursively. For the computation and more information on the partial correlation coefficient, see refs. [44,64].

To provide detailed instruction to our method, we summarize the practical steps here:

Procedure A: MCM for detecting causation from $U = \{u_t\}_{t=1}^L$ to $V = \{v_t\}_{t=1}^L$:

1. Reconstruct the phase space by using delay-coordinate embedding for time series $U$ and $V$, the reconstruction parameters (embedding dimensions $E_u$, $E_v$ and time lags $\tau_u$, $\tau_v$ can be selected by FNN algorithm and by the method of DMI, respectively (see Supplementary Note 5);
2. For each time index $t$, find the set of neighboring points $\mathcal{N}(\mathbf{v}_t)$ of $\mathbf{v}_t$ ($E_v + 1$ nearest neighbors are used since it is the minimum number of points needed for a bounded simplex in an $E_v$-dimensional space[43]);
3. Find the corresponding points in $M_U$ that have the same time indexes as the points in $\mathcal{N}(\mathbf{v}_t)$ and calculate their weighted average (the weights are determined by the distances between the point in $\mathcal{N}(\mathbf{v}_t)$ and $\mathbf{v}_t$, which defines the operation $\mathbb{E}[\cdot]$) to obtain the estimated $\hat{\mathbf{u}}_t^{\mathbf{v}}$;
4. Use an appropriate index (such as $\varrho_C = \left| \text{Corr}(\mathbf{u}_t, \hat{\mathbf{u}}_t^{\mathbf{v}}) \right|$) to characterize the consensus of the estimated time series $\hat{U}^{V_0}$ (subscript 0 is denoted for no translation of $V$ here to keep consistency with the following notations) and the original time series $U$, which measures the causation from $U$ to $V$.

Procedure B: PCM for detecting direct causation from $X$ to $Y$ conditioning on $Z$:

1. Translate time series $Y$ with different candidate time delays $\tau_i (i = 1, 2, \ldots, m)$ to generate $Y_{\tau_i} = \{ y_{t+\tau_i} \}$;
2. For each pair $Z$ to $Y_{\tau_i}$, perform Procedure A to obtain $\text{Corr}(Z, \hat{Z}^{Y_{\tau_i}})$, and denote $\hat{Z}^Y$ as $\hat{Z}^{Y_{\tau_{i_1}}}$, where the time delay $\tau_{i_1}$ maximizes $\text{Corr}(Z, \hat{Z}^{Y_{\tau_i}})$ as in (10);
3. Translate time series $\hat{Z}^Y$ with different candidate time delays $\tau_i (i = 1, 2, \ldots, m)$ to generate $\hat{Z}^Y_{\tau_i}$;
4. For each pair $X$ to $\hat{Z}^Y_{\tau_i}$, perform Procedure A to obtain $\text{Corr}(X, \hat{X}^{\hat{Z}^Y_{\tau_i}})$, and denote $\hat{X}^{\hat{Z}^Y}$ as $\hat{X}^{\hat{Z}^Y_{\tau_{i_2}}}$, where the time delay $\tau_{i_2}$ maximizes $\text{Corr}(X, \hat{X}^{\hat{Z}^Y_{\tau_i}})$ as in (11);
5. For each pair $X$ to $Y_{\tau_i}$, perform Procedure A to obtain $\text{Corr}(X, \hat{X}^{Y_{\tau_i}})$, and denote $\hat{X}^Y$ as $\hat{X}^{Y_{\tau_{i_3}}}$, where the time delay $\tau_{i_3}$ maximizes $\text{Corr}(X, \hat{X}^{Y_{\tau_i}})$ as in (12);
6. Use $\varrho_D = \left| \text{Pcc}(X, \hat{X}^Y | \hat{X}^{\hat{Z}^Y}) \right|$ to measure the direct causation from $X$ to $Y$ conditioning on $Z$.

Note that we search for the strongest causation on different candidate time delays in every MCM procedure above. For consistency, in the whole research, all the MCM results are also based on this strategy. Moreover, it is possible to characterize the causal relations among variables on a distribution of time delays (i.e., a causal spectrum). This full causal description will be included in our future work.

As described above, the first-order PCM method can be established as following definition for networked systems of more than three interacting variables: $X$, $Y$, $Z^1$, ..., $Z^s (s \geq 2)$ (e.g., Fig. 1d), based on which high-order method can be derived,

$$\varrho_{D_1} = \left| \text{Pcc}\left( X, \hat{X}^Y \middle| \left\{ \hat{X}^{\hat{Z}^{iY}} \middle| i = 1, \ldots, s \right\} \right) \right|. \tag{16}$$

In a complex dynamical networks, the indirect causation could also be transferred through more than one variables (e.g., through two variables $X \to Z^1 \to Z^2 \to Y$). The high-order PCM method is derived to specifically characterize this situation. In particular, we calculate the correlation coefficient between $X$ and $\hat{X}^Y$, and the partial correlation coefficient between them through removal of the information about the cross mapping variables via two variables out of the $s$ variables $Z^1$, ..., $Z^s$. The partial correlation coefficient

$$\varrho_{D_2} = \left| \text{Pcc}\left( X, \hat{X}^Y \middle| \left\{ \hat{X}^{\hat{Z}^{z j | Y}} \middle| i \neq j, \, i, j \in \{ 1, \ldots, s \} \right\} \right) \right| \tag{17}$$

represents effectively a second-order method for differentiating the direct and indirect causal links from $X$ to $Y$ that is transferred through two mediate variables. Analogously, the $n$th order measure, denoted by $\varrho_{D_n}$, can be defined through any combinations of $n$ mediate variables from $Z^1$, ..., $Z^s$ as

$$\varrho_{D_n} = \left| \text{Pcc}\left( X, \hat{X}^Y \middle| \left\{ \hat{X}^{\hat{Z}^{i_1 \cdots i_n | Y}} \middle| (i_1, \ldots, i_n) \text{ is an } n - \text{combination from} \{ 1, \ldots, s \} \right\} \right) \right|. \tag{18}$$

Together with $\varrho_C$, $\varrho_{D_n}$ ($n = 1, \ldots, s$) and the PCM measure

$$\gamma = (\Pi_{n=1}^s \varrho_{D_n}) / \varrho_C^s, \tag{19}$$

reflecting the proximity of all these coefficients, we obtain higher-order PCM methods for detecting direct causal links in large networks. However, for a relatively large order $n$, the possible number of combinations of $n$ mediate variables is quite large. We will study the computations and applications of the high-order methods in future work, and in this research, we only consider the first-order problem.

In practice, the partial correlation procedure will encounter calculation problems if the network scale is relatively large and thus a large conditioning set should be taken into account. In this case, we could adopt the technique of selecting several nodes $Z^i$ that maximize $\varrho_C^{X \to Z^i} + \varrho_C^{Z^i \to Y}$ (or $\min\{ \varrho_C^{X \to Z^i}, \varrho_C^{Z^i \to Y} \}$), which means a high probability of the existence of an indirect link through $Z^i$, and

make conditioning on these nodes. Moreover, if we have a priori knowledge that the network is sparse, that is, indirect connections are seldom, we could also make conditioning on $Z^1$, ..., $Z^s$ one by one, and take the minimum value of $\varrho_D^{X \to Y | Z^i}$ as the final result.

Moreover, the PCM idea can be further developed or varied by substituting the partial correlation to other possible measures characterizing the conditional dependence. For example, the coefficient of determination (denoted $r^2$) is a possible choice to serve as an index directly estimated from the cross map neighbors in parceling out effect sizes for each contributing factor. Another heuristic thinking is that for indirect causal influence $X \to Z \to Y$, cutting off either the link $X \to Z$ or $Z \to Y$ is enough to eliminate the whole indirect information flow, which also provides variation of the PCM framework. These further variations will be included in our future work.

## Data availability
The data sets generated during and/or analyzed during the current study are all available from the corresponding author on reasonable request. The links/references for the public data sets used and analyzed during the current study are all provided in Supplementary Information.

## Code availability
The codes as well as their directions for the PCM framework that we developed in this article are publicly available at https://github.com/Partial-Cross-Mapping.

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

## Acknowledgements

W.L. is supported by the National Key R&D Program of China (No. 2018YFC0116600), by the National Natural Science Foundation of China (Nos 11925103 and 61773125), and by the STCSM (Nos 18DZ1201000, 19511132000, and 2018SHZDZX01). L.N.C. is supported by the National Key R&D Program of China (No. 2017YFA0505500), by the Strategic Priority Project of CAS (No. XDB38000000), by the Natural Science Foundation of China (Nos 31771476 and 31930022), and by Shanghai Municipal Science and Technology Major Project (No. 2017SHZDZX01). S.Y.L. and K.A. are supported by JSPS KAKENHI (No. JP15H05707) and by AMED (No. JP20dm0307009). Y.-C.L. is supported by ONR (No. N00014-16-1-2828). H.F.M. is supported by the National Key R&D Program of China (No. 2018YFA0801100) and the National Natural Science Foundation of China (No. 11771010). J.K. is supported by the project RF Government Grant 075-15-2019-1885.

## Author contributions

W.L. and L.N.C. conceived the idea; S.Y.L., H.F.M., W.L., K.A., and L.N.C designed the research; S.Y.L., H.F.M., and W.L. performed the research; All authors, S.Y.L., H.F.M., J.K., Y.-C.L., W.L., K.A., and L.N.C., analyzed the data and wrote the paper.

## Competing interests

The authors declare no competing interests.
