## [Peer Review File · Nature Communications]

Editorial Note: This manuscript has been previously reviewed at another journal that is not operating a transparent peer review scheme. This document only contains reviewer comments and rebuttal letters for versions considered at Nature Communications .

Reviewers' comments:

Reviewer #3 (Remarks to the Author):

The manuscript discusses a very important question of detecting causality from data and addresses the problem of misidentifying indirect causal influences as direct ones. This is a very important question relevant for many applications, and is of interest to those who are developing such methods as well as wider – almost in any area of research where cause-consequence relations in dynamical systems are considered.

The authors propose an interesting and novel method that could help advance the field. While this referee agrees with the authors that "the methods based on prediction frameworks, such as the Granger causality, the transfer entropy, and all their extensions, mathematically are not suitable for dealing with the time series data produced by nonlinear dynamical systems...", their claims of universality of their method may be slightly stretched. Here are the reasons why this may be so –

1. The method is based on the embedding theorem, proposed independently by Takens and Mane (authors may consider adding Mane's reference as well: R. Mañé (1981). "On the dimension of the compact invariant sets of certain nonlinear maps". In D. A. Rand and L.-S. Young (ed.). *Dynamical Systems and Turbulence*, Lecture Notes in Mathematics, vol. 898. Springer-Verlag. pp. 230–242). The theorem is applicable for autonomous systems, while it has been shown in several studies that it has drawbacks when applied to data from nonautonomous systems (see e.g. Clemson and Stefanovska, *Physics Reports*, 2014, DOI: 10.1016/j.physrep.2014.04.001 for discussion). Because most of the real-world systems are nonautonomous, authors could discuss the limitations of their method, or how it can be applicable to nonautonomous systems.
2. In their comparison authors focus on Granger causality and entropy methods, but there are also other methods that have been proposed to study couplings, such as those based on dynamical Bayesian inference (see for example Stankovski et al, *Coupling functions in networks of oscillators*, *New Journal of Physics*, 17: 035002, 2015, or the recent review Stankovski et al *Rev. Mod. Phys.* 89, 2017). To make the claim of universality stronger authors could include comparison/discussion of this type of methods as well.
3. The effect of noise and the number of points. This is a very important issue, but seems insufficiently elaborated in the current version of the manuscript, and again makes the claim of universality weaker. Here, it would be helpful if authors use units of time for their time series (both numerically generated or obtained from measurements), otherwise they can use the term points series instead. Namely, both the effect of noise and the effect of length of the recordings are relative to the dynamical properties of the system in question, hence the usage of absolute time units would help evaluate the effect of noise and the length of recording much more realistically. This referee suggests that all time series are presented in time-units and the discussion about the effect of noise and length of recording is made relative to the time-scales of the system.

Additional comments

1. Authors may consider bringing parts of the supplementary material to the main text. Namely, one of the important part of the work is the concept of non-separability and it is explained in the supplementary material, while it naturally belongs to the main text. The procedure is also central part of the work, and again, is provided in the supplementary material. The examples could also be added to the main text, as, once sufficiently elaborated, they will strengthen the claim of the universality of the method.
2. "Takens' embedding theory" is more specifically the Takens-Mane theorem.

3. The references are relatively well formatted, however many journal titles are written with small letters and abbreviations are not consistently used.
4. Authors may consider making the code publicly available, as well as the data from examples that will be included, to increase the transparency of their work and make possible for others to reproduce it.

Aneta Stefanovska

Reply to Reviewer #3

The manuscript discusses a very important question of detecting causality from data and addresses the problem of misidentifying indirect causal influences as direct ones. This is a very important question relevant for many applications, and is of interest to those who are developing such methods as well as wider – almost in any area of research where cause-consequence relations in dynamical systems are considered. The authors propose an interesting and novel method that could help advance the field. While this referee agrees with the authors that "the methods based on prediction frameworks, such as the Granger causality, the transfer entropy, and all their extensions, mathematically are not suitable for dealing with the time series data produced by nonlinear dynamical systems...", their claims of universality of their method may be slightly stretched. Here are the reasons why this may be so –

Reply. First of all, we thank the reviewer very much for giving us a positive recognition on the broad interest and the scientific relevance of this work. We also thank the reviewer for providing us valuable comments and suggestions which has enabled us to improve our work substantially. We agree with the reviewer that the universality of the proposed method should have a careful and more detailed explanation.

We established a new approach dealing with the situation for which the existing frameworks cannot work effectively. Particularly, our approach, on the one hand, complements the methodology of the well-known CCM framework, and, on the other hand, solves the problems that cannot be efficiently coped by the existing causation detection methods, such as the Granger causality, the transfer entropy, and their conditional versions. We have highlighted clearly the scope of our method and addressed all your concerns in the revised manuscript and in the following responses as well.

1. The method is based on the embedding theorem, proposed independently by Takens and Mane (authors may consider adding Mane's reference as well: R. Mañé (1981). "On the dimension of the compact invariant sets of certain nonlinear maps". In D. A. Rand and L.-S. Young (ed.). *Dynamical Systems and Turbulence*, Lecture Notes in Mathematics, vol. 898. Springer-Verlag. pp. 230–242). The theorem is applicable for autonomous systems, while it has been shown in several studies that it has drawbacks when applied to data from nonautonomous systems (see e.g. Clemson and Stefanovska, *Physics Reports*, 2014, DOI: 10.1016/j.physrep.2014.04.001 for discussion). Because most of the real-world systems are nonautonomous, authors could discuss the limitations of their method, or how it can be applicable to nonautonomous systems.

Reply. We thank the reviewer for your important information on the Takens-Mañé

theorem. We revised the name of the theorem and accordingly cited Mañé's work in the revised manuscript as well.

We fully agree with the reviewer's comment that the Takens-Mañé theorem is applicable only for autonomous systems. So, theoretically, our approach is applicable for autonomous systems only, which is the limitation of our work. However, for some non-autonomous systems with switching structures, our approach is still effective in detecting causation in each switching duration where the structures are invariant. Also, our approach is practically useful for some non-autonomous systems having very slowly changing parameters. Additionally, for some systems whose non-autonomy is induced by external forces or small/moderate noises, our approach could be still practically effective, since there are several versions of the embedding theorem for forced systems, i.e., the versions proposed by Stark [1, 2], supporting the soundness of our approach. For another kind of time-evolving coupled oscillators, the dynamical Bayesian inference provided practical solutions [3]. To be candid, for those non-autonomous systems with very rapidly changing parameters, we still lack very practical method for causation detection and thus are working along this direction presently (also as one of our future topics). We included the above comments clearly in the revision, and listed the main revisions as follows for your quick reference.

“... We emphasize here that strongly coupled (synchronized) variables in nonlinear systems are not in the scope of the PCM framework, because in this circumstance the complete system collapses to the cause system sub-manifold, and the effect variable becomes an observation function on the cause system, where bidirectional causation will always be computationally detected [16]. In addition, theoretically our PCM framework is based on the Takens-Mañé theorem, which is applicable only for autonomous systems. Data entirely recorded from nonautonomous systems are therefore not directly suitable for this framework [42], but our method can be applied to some nonautonomous systems. In particular, it can be used to detect causations with data from switching systems where each duration between the consecutive switching points is sufficiently long, or from systems whose time-variant structures are very slowly changing. Also, our framework is suitable for some forced systems or/and systems with weak or moderate noise because some generalized embedding theorems could support the soundness of our framework [43, 44]. As for another kind of nonautonomous system, i.e., dynamical oscillators with time-evolving coupling functions, the dynamical Bayesian inference using a proper function basis set can provide practical solutions [13]. However, the causation detection in general nonautonomous systems without knowing the explicit model equations but with very rapidly changing parameters and structures remains an open problem and becomes one of our future research topics. ...”

--- from the revision “Results – Partial cross mapping”

2. In their comparison authors focus on Granger causality and entropy methods, but

there are also other methods that have been proposed to study couplings, such as those based on dynamical Bayesian inference (see for example Stankovski et al, Coupling functions in networks of oscillators, New Journal of Physics, 17: 035002, 2015, or the recent review Stankovski et al Rev. Mod. Phys. 89, 2017). To make the claim of universality stronger authors could include comparison/discussion of this type of methods as well.

Reply. Thank you for this constructive comment. We read the related important works suggested by the reviewer, and found that the dynamical Bayesian inference (DBI) is a useful technique to detect couplings in interacting dynamical systems [3, 4]. This method facilitates a comprehensive reconstruction of the dynamical properties of the interacting systems, while the Granger causality-based and transfer entropy-based methods infer only statistical effects. This method shows a powerful effectiveness to infer couplings of the underlying systems, which could be regarded as a mutual complement to our PCM framework. For example, our PCM framework can first detect the basic network structures based on the observed data, significantly simplifying initial regression structure using the function basis set. We cited this work in the main text, and also included a review of this work as well as a comparison study in the SI.

“... We also provided a comparison study between the PCM framework and the dynamical Bayesian inference in SI Section IV. Both methods have their own particular advantages and could be used in a complementary manner. ...”

--- from the revision “Results - Ascertaining direct causation in benchmark systems”

“... First, the method of the dynamical Bayesian inference (DBI) was developed based on an appropriate selection or a priori knowledge of a basis set in some function space for data regression [8-10]. It is applicable for systems even with time-varying couplings. Our PCM framework is a model-free method, only based on the embedding theorem, which is theoretically suitable for dealing with autonomous systems or/and nonautonomous systems with some particular forms as mentioned in the main text. Second, the DBI method can infer the exact coupling functions, while our PCM framework focuses on the detection of causal directions. Last but not least, our PCM is able to distinguish direct causations from indirect ones, while the DBI method probably could be further extended to a conditional version. Both methods have their own particular advantages and could be used in a complementary manner. For example, our PCM framework can first detect the basic network structures based on the observed data, significantly simplifying initial regression structure with the function basis set. We summarize the main differences between these methods in Table S5. ...”

--- from the revised SI Section IV.C

TABLE S5. Comparison of the PCM framework with the DBI method.

	PCM	DBI
Not require, a priori , knowledge of model equations	✓	×
Infer coupling functions using function basis set	×	✓
Distinguish direct couplings from indirect ones	✓	×
Infer structures with slowly-switched durations	✓	✓

3. The effect of noise and the number of points. This is a very important issue, but seems insufficiently elaborated in the current version of the manuscript, and again makes the claim of universality weaker. Here, it would be helpful if authors use units of time for their time series (both numerically generated or obtained from measurements), otherwise they can use the term points series instead. Namely, both the effect of noise and the effect of length of the recordings are relative to the dynamical properties of the system in question, hence the usage of absolute time units would help evaluate the effect of noise and the length of recording much more realistically. This referee suggests that all time series are presented in time-units and the discussion about the effect of noise and length of recording is made relative to the time-scales of the system.

Reply. We agree with the reviewer that the absolute time units should be used in characterizing the dynamical properties of the underlying systems, because the considered number of points actually depends on the sampling rate, and thus cannot represent the real time span independently. In light of the reviewer’s suggestion, we revised all the descriptions in the manuscript, presenting time series in their real time units and describing clearly the sampling rate/time unit. More precisely, we use the logistic map as a benchmark model, which was once regarded as a discrete-time demographic model characterizing the population changes. In this circumstance, the discrete-time unit matches exactly the time unit of the parameters, which could be second, day, month, or even year. For the other systems studied in our work, we highlighted the sampling rates and the real time units of the time series. Moreover, the effect of noise is characterized by its scale relative to the system instead of the absolute strength of the noise. We list the related revision below for your quick reference.

“... Different lengths of time series (i.e., 100,200,500,1000,3000) and different noise levels (i.e., 0,0.005,0.01,0.015,0.02) are, respectively, taken into account. Here, the sampling rate is 1Hz, so the number of time points matches exactly the time unit of the system. We use the term “time length” to denote the absolute length of the time duration for a time series. Additionally, we generate the white noises with respect to each time unit and use the term “noise scale” as the ratio between the noise strength and the amplitude of the system dynamics (e.g., the

amplitude is 1 for the logistic map, so the noise strength is exactly the same as the noise scale here). ...”

--- from the revised SI Section IV.A

To further demonstrate the robustness of our method, which is a very important issue as stated by the reviewer, we performed an additional test on the robustness of our PCM framework against the time series lengths and the noise scales for the networked benchmark model consisting of eight species. The numerical results showed that, for this model, our framework is effective even with small data size and relatively strong noise. We thank the reviewer’s suggestion, which rendered our work more integrated. We list the results of the new test here for your quick reference.

“... To further explore the robustness of the PCM framework against time series lengths and the noise scales, we performed an additional numerical analysis using the eight species system introduced in Section III. As expected, Figure S9 shows that the detection accuracy increases with the time length of the time series used in simulations but always remains at a high level, confirming the effectiveness of our PCM framework in dealing with an extremely small amount of data. In addition, increasing the noise scale only slightly lowers the detection accuracy. Note that the system becomes divergent as the noise scale is larger than 0.01 [shown by “×” on the horizontal axis in Fig. S9(b)]. These also demonstrate that our PCM framework is useful when the noisy perturbation is introduced in a manner of small or moderate intensity. ...”

--- from the revised SI Section IV.A

FIG. S9. Robustness tests of PCM against time series of different time lengths and noise scales with the networked system consisting of eight species in Section III. The shaded area represents the standard deviation of the PCM results of 100 simulations. The system becomes divergent when the noise scale exceeds 0.01 (shown by “×” on the horizontal axis). Here the time lengths denote the system’s absolute time unit and the noise scales are the ratio of the noise strength to the system’s amplitude. The threshold value is selected to be 0.5 in this analysis.

Additional comments

1. Authors may consider bringing parts of the supplementary material to the main

text. Namely, one of the important part of the work is the concept of non-separability and it is explained in the supplementary material, while it naturally belongs to the main text. The procedure is also central part of the work, and again, is provided in the supplementary material. The examples could also be added to the main text, as, once sufficiently elaborated, they will strengthen the claim of the universality of the method.

Reply. Because the “Introduction, Results, Discussion” sections have a limit of 5000 words and the “Method” section has no word-limit, we have reorganized the whole manuscript to incorporate several crucial parts describing the method into the main text “Method” section according to the above comment. And all the additional information regarding to the examples is fully discussed in the main text with only the technical details (with clear guidance) left in the SI.

2. “Takens’ embedding theory” is more specifically the Takens-Mane theorem.

Reply. Thanks for the reviewer’s important information on this theorem. We have made the correction, fully changing from the “Takens theorem” to the “Takens-Mañé theorem” in the revised manuscript.

3. The references are relatively well formatted, however many journal titles are written with small letters and abbreviations are not consistently used.

Reply. We have carefully checked and revised the information of the references, and amended all those improper formats used in the original manuscript.

4. Authors may consider making the code publicly available, as well as the data from examples that will be included, to increase the transparency of their work and make possible for others to reproduce it.

Reply. Many thanks for the reviewer’s suggestion. We have provided necessary references and clear access to all data sets used in this research and made the codes (and data sets without download links) publicly available at <https://github.com/Partial-Cross-Mapping>.

We would like to thank again the reviewer for the tremendous effort and precious time to review our work, providing us with very insightful comments and suggestions which made our work improved substantially.

References

- [1] Stark, J. (1999). Delay embeddings for forced systems. I. Deterministic forcing. *Journal of Nonlinear Science*, 9(3), 255-332.
- [2] Stark, J., Broomhead, D. S., Davies, M. E., & Huke, J. (2003). Delay embeddings for forced systems. II. Stochastic forcing. *Journal of Nonlinear Science*, 13(6), 519-577.
- [3] Stankovski, T., Ticcinelli, V., McClintock, P. V., & Stefanovska, A. (2015). Coupling functions in networks of oscillators. *New Journal of Physics*, 17(3), 035002.
- [4] Stankovski, T., Pereira, T., McClintock, P. V., & Stefanovska, A. (2017). Coupling functions: universal insights into dynamical interaction mechanisms. *Reviews of Modern Physics*, 89(4), 045001.

Reviewers' comments:

Reviewer #3 (Remarks to the Author):

The authors have substantially improved the manuscript. However, there are several remaining points that need to be addressed before this referee can recommend the paper for publication.

1. Universality. The authors have made a great effort to indicate the originality of the approach they introduce. However, it seems that they have omitted discussion on some of the previous works on the problem of direct-indirect interactions, which is quite a well-studied topic. For example, works like

- i. Schelter, Winterhalder, Hellwig, Guschlbauer, Lücking, and Timmer, Direct or indirect? Graphical models for neural oscillators, *Journal of Physiology-Paris*(2006), 99 (1), 37-46
- ii. Nawrath, Romano, Thiel, Kiss, Wickramasinghe, Timmer, Kurths, and Schelter, Distinguishing Direct from Indirect Interactions in Oscillatory Networks with Multiple Time Scales, *Phys Rev Lett* (2010), 104, 038701, 10.1103/PhysRevLett.104.038701

are quite relevant to the work presented in the manuscript, and the authors should carefully consider and discuss the novelty of their approach in comparison to these earlier ideas.

2. Comparison with the DBI. In general, it is easy to agree that the proposed PCM and DBI are complementary. However, the comparison and the arguments in Table S5 are rather 'cherry-picking' arguments on the side of PCM. Looking at the table in detail -

- i. 'Not require, a priori, knowledge of model equations' - this is true in general. The DBI does not require specific knowledge for oscillators (as the model is based on Fourier series), yet, in general, it is a model-based method.
- ii. 'Infer coupling functions using function basis set' - indeed; yet, not only coupling functions but also other dynamical mechanisms too. In this respect the PCM is a 'functional connectivity' while DBI is a 'effective connectivity' measure [see e.g. Park& Friston (2013) *Science*, 342(6158), 1238411]
- iii. 'Distinguish direct couplings from indirect ones', true. Though DBI is not designed for this problem, this is the complementary aspect.
- iv. Here one very important aspect is left out - the DBI is a stochastic measure and can infer the noise dynamics, while PCM is not.
- v. 'Infer structures with slowly-switched durations' - this is not true for PCM in general. The slowly-switched duration is a relative term (how slow? in respect of what?), in this respect. During the switching from moderate to fast the PCM will not work by definition, as it is based on the Takens-Mane theorem. DBI will perform quite satisfactorily in such cases by definition (see e.g. fig5 in Duggento et al, *PRE* 86(6), 061126, 2012).

So, either delete this argument all-together, or have two rows/arguments where for slow (but define what slow means) both PCM and DBI are fine, and one for moderate and fast where only DBI is fine.

3. The writing style. There is some degree of aggression in the writing style - especially in the main document (though not in the response letter, at all), and the authors may consider editing it to smooth it down.

Aneta Stefanovska

Reply to Reviewer #3

The authors have substantially improved the manuscript. However, there are several remaining points that need to be addressed before this referee can recommend the paper for publication.

Reply. We thank the reviewer for your positive recognition on our previous revised manuscript. We are also grateful for your further valuable comments, which are quite important for improving the completeness of this work. We have addressed properly all your concerns in the further revised manuscript and in the following responses as well.

1. Universality. The authors have made a great effort to indicate the originality of the approach they introduce. However, it seems that they have omitted discussion on some of the previous works on the problem of direct-indirect interactions, which is quite a well-studied topic. For example, works like

(i). Schelter, Winterhalder, Hellwig, Guschlbauer, Lücking, and Timmer, Direct or indirect? Graphical models for neural oscillators, *Journal of Physiology-Paris* (2006), 99 (1), 37-46

(ii). Nawrath, Romano, Thiel, Kiss, Wickramasinghe, Timmer, Kurths, and Schelter, Distinguishing Direct from Indirect Interactions in Oscillatory Networks with Multiple Time Scales, *Phys Rev Lett* (2010), 104, 038701, 10.1103/PhysRevLett.104.038701

are quite relevant to the work presented in the manuscript, and the authors should carefully consider and discuss the novelty of their approach in comparison to these earlier ideas.

Reply. We thank the reviewer for providing information on these previous important works. We have carefully read these works and also reviewed the other relevant works on the topic of distinguishing the direct from the indirect interactions in particular dynamical systems. We find that most of these interesting works [including the above Ref. (i)], combining the methods/frameworks of the Granger causality and its extensions, the transfer entropy and its extensions, and the other related measures with the graphical models, provided a visible and comprehensive description of the causal relations among interested variables. However, these works still fall into the category of methods that require the separability property which is not always valid for the variables in nonlinear dynamical systems. Also, we find that some methods, e.g., the partial recurrence based on synchronization [see the above Ref. (ii)], can distinguish the direct from the indirect links; however, these links are not directional, so they cannot be regarded as causation but association only. We highly appreciate all the significant advances in causation detection techniques, and also hope that our method could be used to address the problem of causation detection in complex dynamical systems where the variables are non-separable. Accordingly, we make the revisions and cite the useful references. We list the revisions as follows

for your quick reference.

*“There were previous **studies of significant advance** in detecting direct causal links to reconstruct the underlying true causal network based on the concept of partial transfer entropy or its linear Gaussian version, the conditional Granger causality, which resulted in many successful data mining in related fields [32-38]. Combining these methods with graphical models, recent studies further provided a visible and comprehensive description of causal relations among interested variables [36,38,39]. However, mathematically, **all these** methods are not applicable directly in situations where the relevant dynamical variables are non-separable so that the “information” from any variables cannot be separated easily in a prediction framework (see Method for the rigorous concept of non-separability). In real-world nonlinear systems, the non-separability is ubiquitously present among systems variables. To our knowledge, the problem of ascertaining direct causation by removing indirect causal influences for general complex dynamical systems **has not been fully studied and** remained outstanding.”*

2. Comparison with the DBI. In general, it is easy to agree that the proposed PCM and DBI are complementary. However, the comparison and the arguments in Table S5 are rather 'cherry-picking' arguments on the side of PCM. Looking at the table in detail -

- i. 'Not require, a priori, knowledge of model equations' - this is true in general. The DBI does not require specific knowledge for oscillators (as the model is based on Fourier series), yet, in general, it is a model-based method.
- ii. 'Infer coupling functions using function basis set' - indeed; yet, not only coupling functions but also other dynamical mechanisms too. In this respect the PCM is a 'functional connectivity' while DBI is an 'effective connectivity' measure [see e.g. Park & Friston (2013) Science, 342(6158), 1238411]
- iii. 'Distinguish direct couplings from indirect ones', true. Though DBI is not designed for this problem, this is the complementary aspect.
- iv. Here one very important aspect is left out - the DBI is a stochastic measure and can infer the noise dynamics, while PCM is not.
- v. 'Infer structures with slowly-switched durations' - this is not true for PCM in general. The slowly-switched duration is a relative term (how slow? in respect of what?), in this respect. During the switching from moderate to fast the PCM will not work by definition, as it is based on the Takens-Mane theorem. DBI will perform quite satisfactorily in such cases by definition (see e.g. fig5 in Duggento et al, PRE 86(6), 061126, 2012). So, either delete this argument all-together, or have two rows/arguments where for slow (but define what slow means) both PCM and DBI are fine, and one for moderate and fast where only DBI is fine.

Reply. Thanks for the reviewer’s careful explanations and discussions on the comparison of DBI and PCM. We totally agree with the comments and the suggestions from the reviewer. So, we rewrite the related paragraphs in the main text and in the SI as well, also deleting those inappropriate expressions and the Table

on comparison. We list the related revisions below for your quick reference.

Main Text: *“In particular, it can be numerically used to detect piecewise causations with data from switching systems where the switching points could be located and each duration between the consecutive switching points is sufficiently long. Also, our framework is suitable for some forced systems or/and systems with weak or moderate noise because some generalized embedding theorems could support the soundness of our framework [49,50]. As for an important kind of nonautonomous system, viz., dynamical oscillators with time-evolving coupled functions or/and with various types of noise, the dynamical Bayesian inference with a delicate set of function bases can provide pretty practical solutions [14]. As for the future research topics, possible investigations include combining the above mutually complementary methods for causation detection in more general dynamical systems without knowing explicit model equations but with highly complex interaction structures.”*

SI: *“First, the method of the dynamical Bayesian inference (DBI) does not require the detailed knowledge of the explicit equations in the models but only uses a delicate selection of a basis set in some function space for data regression [8-11]. It is applicable for general autonomous and nonautonomous systems. Our PCM framework is a model-free method, only based on the embedding theorem, which is theoretically suitable for dealing with autonomous systems or/and nonautonomous systems with some particular forms as mentioned in the main text. More concretely, the PCM framework works for the switching systems where the switching points can be located and each duration between the consecutive switching points is sufficiently long, while the DBI is applicable for more types of nonautonomous systems, including the dynamical oscillators with time-evolving coupled functions or/and with various types of noise [11]. Second, the DBI method can infer the exact coupling functions and underlying dynamical mechanisms, while our PCM framework focuses much on the detection of causal relations. So, the connections detected by the DBI could be regarded as effective connectivity while the causal relations found by the PCM framework are more like functional connectivity [12]. Additionally, our PCM is able to distinguish direct causations from indirect ones, while the DBI method could be further extended to a conditional version. Both methods have their own particular advantages and could be used in a complementary manner. For example, in highly complex networks, our PCM framework can first detect the basic network structures based on the observed data, significantly simplifying initial regression structure with the function basis set.”*

3. The writing style. There is some degree of aggression in the writing style - especially in the main document (though not in the response letter, at all), and the authors may consider editing it to smooth it down.

Reply. Thank you very much for this suggestion. We have amended the sentences and expressions to have all the presentations in a more appropriate and smooth manner.

We would like to thank the reviewer again for your precious time and constructive comments and suggestions, which made our work much improved. Also we hope that everything goes well with you during such a season of pandemic COVID-19.

References

- [1] Guo, S., Seth, A. K., Kendrick, K. M., Zhou, C., & Feng, J. (2008). Partial Granger causality—eliminating exogenous inputs and latent variables. *Journal of Neuroscience Methods*, 172(1), 79-93.
- [2] Frenzel, S., & Pompe, B. (2007). Partial mutual information for coupling analysis of multivariate time series. *Physical Review Letters*, 99(20), 204101.
- [3] Zhao, J., Zhou, Y., Zhang, X., & Chen, L. (2016). Part mutual information for quantifying direct associations in networks. *Proceedings of the National Academy of Sciences*, 113(18), 5130-5135.
- [4] Runge, J., Heitzig, J., Petoukhov, V., & Kurths, J. (2012). Escaping the curse of dimensionality in estimating multivariate transfer entropy. *Physical Review Letters*, 108(25), 258701.
- [5] Schelter, B., Winterhalder, M., Hellwig, B., Guschlbauer, B., Lücking, C. H., & Timmer, J. (2006). Direct or indirect? Graphical models for neural oscillators. *Journal of Physiology-Paris*, 99(1), 37-46.
- [6] Nawrath, J., Romano, M. C., Thiel, M., Kiss, I. Z., Wickramasinghe, M., Timmer, J., ... & Schelter, B. (2010). Distinguishing direct from indirect interactions in oscillatory networks with multiple time scales. *Physical Review Letters*, 104(3), 038701.
- [7] Runge, J. (2018). Causal network reconstruction from time series: From theoretical assumptions to practical estimation. *Chaos: An Interdisciplinary Journal of Nonlinear Science*, 28(7), 075310.
- [8] Runge, J., Petoukhov, V., & Kurths, J. (2014). Quantifying the strength and delay of climatic interactions: The ambiguities of cross correlation and a novel measure based on graphical models. *Journal of Climate*, 27(2), 720-739.
- [9] Park, H. J., & Friston, K. (2013). Structural and functional brain networks: from connections to cognition. *Science*, 342(6158), 1238411.
- [10] Duggento, A., Stankovski, T., McClintock, P. V., & Stefanovska, A. (2012). Dynamical Bayesian inference of time-evolving interactions: From a pair of coupled oscillators to networks of oscillators. *Physical Review E*, 86(6), 061126.

REVIEWERS' COMMENTS:

Reviewer #3 (Remarks to the Author):

The authors have addressed all my comments and the paper can now be accepted for publication.